# A Cloud-IoT Architecture for Latency-Aware Localization in Earthquake Early Warning

**DOI:** 10.3390/s23208431

**Published:** 2023-10-13

**Authors:** Paola Pierleoni, Roberto Concetti, Alberto Belli, Lorenzo Palma, Simone Marzorati, Marco Esposito

**Affiliations:** 1Department of Information Engineering (DII), Università Politecnica delle Marche, 60131 Ancona, Italy; p.pierleoni@univpm.it (P.P.); a.belli@univpm.it (A.B.); m.esposito@pm.univpm.it (M.E.); 2Istituto di Istruzione Superiore Carlo Urbani, 63821 Porto Sant’Elpidio, Italy; roberto.concetti@gmail.com; 3Istituto Nazionale di Geofisica e Vulcanologia (INGV), Osservatorio Nazionale Terremoti, 60131 Ancona, Italy; simone.marzorati@ingv.it

**Keywords:** cloud computing, early warning systems, earthquake localization, Internet of Things

## Abstract

An effective earthquake early warning system requires rapid and reliable earthquake source detection. Despite the numerous proposed epicenter localization solutions in recent years, their utilization within the Internet of Things (IoT) framework and integration with IoT-oriented cloud platforms remain underexplored. This paper proposes a complete IoT architecture for earthquake detection, localization, and event notification. The architecture, which has been designed, deployed, and tested on a standard cloud platform, introduces an innovative approach by implementing P-wave “picking” directly on IoT devices, deviating from traditional regional earthquake early warning (EEW) approaches. Pick association, source localization, event declaration, and user notification functionalities are also deployed on the cloud. The cloud integration simplifies the integration of other services in the architecture, such as data storage and device management. Moreover, a localization algorithm based on the hyperbola method is proposed, but here, the time difference of arrival multilateration is applied that is often used in wireless sensor network applications. The results show that the proposed end-to-end architecture is able to provide a quick estimate of the earthquake epicenter location with acceptable errors for an EEW system scenario. Rigorous testing against the standard of reference in Italy for regional EEW showed an overall 3.39 s gain in the system localization speed, thus offering a tangible metric of the efficiency and potential proposed system as an EEW solution.

## 1. Introduction

Earthquake early warning (EEW) systems have been rapidly developing in recent years as a result of the development of information technology. EEW systems are able to analyze ground motions in real time and provide alerts before the onset of the destructive seismic waves at specific targets [1]. These systems are mainly based on the different propagation speeds of the waves generated by an earthquake. When a seismic event occurs, different types of seismic waves propagate from the hypocenter, namely Primary waves (P-waves), Secondary waves (S-waves), and surface waves. The greatest destructive power is contained in the S-waves and in the surface waves. P-waves contain relevant information about the event in progress and do not carry the same destructive power, while also moving at higher speeds. An EEW system must be able to identify the arrival of P-waves and send alerts in case of a potential harmful event. Another important feature of an EEW system is the ability to estimate the position of the earthquake source and the arrival times at a certain target location. The hypocenter and/or epicenter localization problem can be solved by looking at the arrival times, incidence direction, and/or horizontal velocity of seismic waves at a seismic station, using a single-station approach or utilizing multiple stations [2]. In regional EEW systems, a sequence of streams from different stations at a central server or a data center are analyzed at a central hub through a “picking” algorithm that is able to detect P-wave onsets in the seismic trace, also called “picks” in seismological jargon. The detected P-wave picks are then used to estimate the earthquake source. There are different methods that are able to give an estimation of the epicenter location, and the estimation error is usually a function of the number of triggers [3]. Since stations are located at different distances from the epicenter, this also means that the estimation error becomes a function of time. Therefore, a greater concentration of stations in the area for the real-time acquisition of waveforms leads to benefits both in terms of the correctness of results and detection times. Until a few years ago, the densification of seismic networks would have been challenging due to the high costs of dedicated seismic instrumentation. However, recent developments in the Internet of Things (IoT) have paved the way for promising low-cost solutions in a growing number of applications, such as smart homes [4,5,6], smart cities [7,8], renewable resources systems [9,10], and disaster management [11,12].

In this work, a complete cloud-IoT architecture for a regional EEW system is proposed. It integrates a newly developed localization algorithm and it uses the Message Queuing Telemetry Transport (MQTT) protocol for the exchange of picking information between IoT devices and cloud resources. Unlike existing regional EEW and earthquake monitoring networks [13,14], the proposed architecture does not require the continuous streaming of acquired ground motion traces towards a central hub or data center for processing, but it performs the P-wave picking on IoT devices. This implementation choice means that each device in the EEW network only has to send picking information towards the cloud to perform localization or, if needed for other functionalities, just short traces centered around the P-wave pick. This greatly reduces the amount of transmitted information and power consumption, which are essential factors in IoT applications, and it also lowers the latency introduced by centralized processing. An epicenter localization algorithm is also developed in this work. The algorithm is based on the hyperbola method but makes use of Time Difference of Arrival (TDoA). TDoA is a positioning algorithm that is used in many IoT applications, for example, for the geolocalization of radio-frequency emitters such as smart tags, and it is based on arrival times. The inputs of the developed algorithm are, therefore, the pick times of the P-wave at the stations of the seismic/IoT network. These “picks” are used to iteratively build hyperbolas through TDoA, reducing the estimation error at each iteration as more devices detect the P-wave.

Besides testing the accuracy of the localization algorithm, experiments in terms of latency are also carried out to test the suitability of the developed solution for EEW. The performance of the on-board triggering, localization algorithm, and the overall IoT application are compared with the PRobabilistic and Evolutionary early warning SysTem (PRESTo) software [13,15], which represents the Italian and European reference standard for regional EEW [16,17,18], with dedicated software modules for picking, localization, and magnitude estimation.

Overall, the main contributions of the paper are the following:
The introduction of decentralized “picking” of seismic events, bringing the process closer to IoT devices. This decentralization optimizes data latency and enhances the efficiency of event detection;The use of the MQTT protocol for data transmission, which has proved to be a valuable alternative to standards usually employed for seismic data transmissions [19];The design of a complete IoT architecture integrating cloud resources, showcasing a novel and comprehensive system tailored for earthquake detection, localization, and event notification;The development of a localization algorithm, designed to take advantage of the decentralized picking. This algorithm utilizes the hyperbola method and TDoA multilateration and is deployed on a centralized cloud server.

The rest of the paper is organized as follows: Section 2 presents the concepts governing regional EEW and a comparison of state-of-the-art EEW systems in Europe; in addition, examples of recent applications of the IoT to EEW and trends in earthquake localization are explored. Section 3 describes the proposed solution, including the proposed IoT architecture, the application flow of the developed solution, and the localization algorithm. Section 4 analyzes the performance of the algorithm and the overall performance of the developed IoT localization system compared to PRESTo. Section 5 presents conclusions and future developments.

## 2. Related Works

EEW systems can be divided into on-site, regional, and hybrid EEW systems [16,17]. A regional EEW system is used to detect, locate, and determine the magnitude of seismic events analyzing streams from stations belonging to a regional monitoring network [17]. Notable examples of regional EEW systems are ShakeAlert [20], PRESTo [13], and the Virtual Seismologist [21].
PRESTo [15] is a software platform for regional EEW that contains modules for waveform data acquisition from stations using the Seedlink protocol, seismic phase picking using the FilterPicker algorithm, source localization, and peak magnitude estimation [13].

Studies have shown that implementing a PRESTo in Italy could theoretically deliver alert messages within approximately 5 to 10 s, with maximum lead times for implementing security measures at a distant target of about 25 s [16]. This system, therefore, represents the current state-of-the-art in regional EEW technology in the country and is one of the most used software solutions for EEW in Europe [18]. Besides PRESTo, the other most widely used regional EEW software in Europe is VS(SC) (Virtual Seismologist in SeisComP) [21]. VS(SC) incorporates phase picking using the Short-Time Average over Long-Time Average (STA/LTA) algorithm [22], and location and magnitude estimation algorithms integrated in the Seiscomp software module [23]. It has been shown that PRESTo is ultimately the best choice for regional EEW compared to the VS(SC) module [18], as it provides better short-term predictions, a key requirement for EEW. Furthermore, PRESTo has undergone extensive implementation and testing in multiple regions and scenarios, including Southern Italy, Turkey, Romania, Southern Iberia, Austria, and Slovenia [18]. Additionally, the Italian national earthquake monitoring agency (Istituto Nazionale di Geofisica e Vulcanologia, INGV) has conducted tests to assess its performance when integrated with existing monitoring infrastructures across different regions. It has been shown how a PRESTo-based regional EEW system in the same region as the one chosen for the tests that have been carried out in this paper would provide maximum lead times from 5 to 10 s [17]. PRESTo is, therefore, considered as the reference EEW software architecture for the tests carried out in this study [15]. Table 1 presents a comparison of the state-of-the-art regional EEW systems and the solution proposed in this paper. Compared to the state of the art of regional EEW systems, such as PRESTo, this work introduces a distributed approach to picking so as to reduce the latency introduced by waveform streaming and processing at a central hub, thus improving the overall localization and event declaration delays.

In recent years, there have been many advances in the use of the IoT for early warning purposes, including EEW [11], taking advantage of the developments in Micro-Electro-Mechanical System (MEMS) technology, cloud and edge computing, and the crowd-sensing paradigm. A notable example of a mobile crowd-sensing application for EEW is the MyShake application [24], which distributes computing among smartphone devices. Each device running the MyShake app is used for on-site earthquake detection using data from the smartphone-embedded sensors, and then a centralized algorithm is used to distinguish between “true” earthquake events and false triggers. Two notable works that heavily integrate cloud computing in their architectures are the SeismoCloud project [25] and the EarthCloud project [26]. The former is another example of smartphones used to detect seismic events, even though any ground motion measuring device can be integrated in the developed cloud platform, while the latter makes use of IoT devices equipped with geophones that stream data towards an Amazon Web Services (AWS) IoT Core. A warning is issued when a certain number of devices from the same location exceed a set threshold. Similar to this work, the use of cloud computing provides these solutions with enhanced computational capabilities, but also additional services that can be used for device monitoring, reliable data dispatching, data storage, etc.

There are different computer-based approaches for automated earthquake locations that use seismic-phase arrival time observations. Digital computers permit the use of iterative linearized approaches based on Geiger’s method [27], while the increasing power of computing has made possible the use of large-scale, grid and stochastic direct searches with probabilistic nonlinear methods that provide a complete description of location uncertainties [28]. A far simpler manual method for epicenter localization consists in drawing a circle around the stations that were triggered by the earthquake with a radius equal to the epicentral distance, calculated from the P-wave and S-wave arrival times and assuming a constant speed for both waves. The epicenter then lies in a locus given by the intersection of the circles, and it can be roughly determined choosing, for example, the center of gravity of the locus. Graphical methods such as the hyperbola method have become, once again, interesting with the development of EEW systems that require speed, simplicity, and reliability [29]. PRESTo uses the RTLoc algorithm, a derivative of the hyperbola method [3]. Another technique based on arrival times and the hyperbola method is the one developed by Rydelek et al. [30]. This method only requires a subarray of two stations to detect the earthquake source and it is based on the concept of not-yet-arrived data. It is a quick method whose effective time savings depend on the station geometry and the epicentral position. The looser focus on precision is justified by the EEW application scenario. EEW systems are usually employed to detect strong seismic events that might have large fault ruptures, and with the more pressing requirement of acting quickly rather than with high accuracy.

Recently, Machine Learning (ML)-based approaches to localization have also seen a wider use. While some are based on waveforms detected at different stations or even single stations [31,32], others make use of the wave arrival times to estimate the source of the earthquake in a similar way to some of the previously described methods. Convolutional neural networks [33], attention-based networks [34], and other ML methods [35] have shown good accuracy in determining the epicentral/hypocentral position, with different degrees of spatial resolution; however, to the authors’ knowledge, most have either been tested in a vacuum or do not provide figures on their time performance, with the exception of [35], which provides some estimated figures on its implementation in an EW scenario.

Studies have demonstrated the feasibility of utilizing TDoA for earthquake localization [36,37,38]. However, these works do not evaluate the application of their method in the EEW scenario and focus on enhancing focal parameter estimation compared to state-of-the-art estimation [36], or, more generally, adopt complex 3D models [38], prioritizing accuracy and/or robustness over timeliness.

Table 2 presents a comparison of the main works related to the solution proposed in the paper. The table highlights the following features of related papers that have been cited in this work: main focus of the paper/project, employed/proposed localization strategy (if any), employed cloud platform (if any), main novelty introduced by the paper/project. Compared to other works on earthquake localization, this study evaluates the whole architecture of the proposed earthquake early warning system, from picking to localization to user notification, and evaluates the efficiency of the developed algorithm not only in terms of accuracy, but also in terms of latency and overall performance when integrating the developed solution in an end-to-end warning system. As accuracy is not a main constraint in EEW, accurate estimations of other parameters, such as focal parameters, that can be performed during the early alert were not considered.

## 3. Proposed Solution

In this work, a complete cloud-IoT architecture for seismic event declaration and earthquake localization is proposed. The architecture integrates cloud resources and a new localization algorithm that is developed and tested with real waveforms. The MQTT protocol is used to exchange information between IoT devices and the cloud. The choice of MQTT is justified by a recent work by the authors showing that it has a better performance in early warning scenarios compared to other communication protocols commonly used in seismic data exchange, namely the Seedlink protocol [40], with an average gain of more than 600 ms to transmit miniSEED packets [19]. It is also a very flexible protocol, and simple middleware services can be implemented to allow communication between MQTT-based architectures and “classic” Seedlink-based seismic monitoring networks. Furthermore, MQTT enables an easy integration of cloud resources into IoT applications since it is the most commonly used middleware service in commercial cloud platforms for the IoT, such as AWS, Microsoft Azure, or Google Cloud Platform [41].

The AWS cloud platform is chosen as the cloud provider, having shown good performance for several key indicators for Internet of Things applications [41].

The proposed IoT architecture has the following functionalities:Sampling acceleration values and performing the picking of the P-wave on board of an IoT device;Sending a message to an MQTT broker deployed on the cloud when a P-wave is detected;Checking the temporal consistency of triggers received on the cloud;Performing event localization when a predefined number of consistent triggers is received on the cloud from IoT devices;Evaluating the arrival times of the S-wave at a target location;Sending an alert message to users via the AWS service called the Simple Notification Service (SNS) if a certain Peak Ground Acceleration (PGA) threshold is met.

Furthermore, the cloud integration and the MQTT protocol allow for the seamless integration of additional services, such as device monitoring and storage services. The proposed architecture is designed to shift the processing closer to the data source, moving the seismic wave picking on the IoT devices themselves. Additionally, services like AWS Greengrass let users deploy cloud resources in proximity to the edge or the IoT device itself, further enhancing the efficiency of the cloud/edge interaction. While with the current, rather sparse seismic networks, a fully centralized approach is required to perform localization, with a more densely deployed seismic network, a multiaccess edge computing approach could be a valuable tool to elaborate the triggers from local stations and perform early localization. In conjunction with 5G reliability, network slicing, and services such as uRLLC, this could provide earlier localization by also elaborating triggers closer to the device, enhancing reliability and overall performance [42]. The rest of the section provides details about the IoT architecture, the application flow from earthquake detection to localization to event declaration, and the developed localization algorithm.

### 3.1. Proposed Internet of Things Architecture

A cloud-IoT architecture that integrates a software module for fast event detection and source estimation is proposed. The architecture consists of three layers: (i) A perception layer to measure ground acceleration, detect P-waves, and send pick notifications to an MQTT broker; (ii) an application layer with a software module that checks the temporal consistency of received picks, performs the localization, declares the event, and sends a notification to higher layers through the Amazon SNS service; (iii) an enterprise layer to offer applications, interfaces, and notification services to users. The AWS integration simplifies the deployment of other, additional functionalities in each layer.

In detail, the perception layer consists of an IoT device that measures acceleration, detects P-waves, and sends the P-wave arrival times and accelerometer measurements to an MQTT broker. In this paper, the IoT device is considered to be a Single-Board Computer (SBC) equipped with an accelerometer. MEMS accelerometers can be used for strong motion detection purposes, and, in a previous work by the authors, it was shown that an SBC such as a Raspberry Pi can be used for data acquisition, transmission, integration of the picking algorithms [43], and other functionalities such as Seedlink servers. There are many algorithms for P-wave picking, and the FilterPicker is chosen for this work, as it requires minimal resources [44] and is used by the earthquake detection module of PRESTo. Unlike PRESTo, in the proposed architecture, P-wave picking is performed on board of the device and, once a pick is detected, the device immediately publishes a message on a dedicated topic (in this case, /deviceID/picks) on an MQTT broker in the application layer. The MQTT message contains the P-wave detection timestamp and the P-wave pick timestamp. This implementation choice is used to avoid the data latency introduced by the station dataloggers and the transmission of accelerometric traces to a central processing hub for picking, as is the usual method in regional EEW systems. Alongside the pick message, a short portion of the waveform is transmitted to the broker on a dedicated topic for the estimation of the event’s PGA value on the application layer. Additionally, if needed and if they can afford it in terms of memory and processing power, perceptionlayer devices can also be configured as Seedlink servers [40] to send continuous streams in miniSEED format to external clients [43].

The application layer consists of an MQTT broker that receives pick notifications from the perception layer devices, and forwards them to a software module that performs the localization when it receives a certain amount of consistent triggers. Supplementary functionalities beyond localization and event notification can be deployed on the application layer, such as Seedlink servers and clients for the real-time data transmission of traces, data storage (via the sl-archive software [45], and/or Amazon S3), device-monitoring services, and user notifications.

The software module developed for EEW performs the following operations:Subscription through wildcard to the topic +/picks;Subscription through wildcard to the topic +/trace;Association of the arrival times of the P phases when a configurable amount of picks is received from the +/picks topic;Localization of the epicenter;Computation of the PGA values from the data published on +/trace by the IoT devices;Estimation of the arrival time of S-waves at one or more target points;Event notification through a message on the /earthquake topic.

The last three operations are performed by the event declaration submodule. The above application flow is also illustrated in Figure 1.

The minimum number of triggers needed for the localization can be configured by the users, and another configurable threshold tells the algorithm the number of triggers before it can stop the estimation. By employing a multitrigger approach, the algorithm becomes more resistant to false picking [35]. When a message is received containing the P phase detected by a device, the software stores it in a list and determines its consistency with the propagation of an earthquake with respect to other stored picks. The consistency is verified with the following time coincidence criterion: considering an average speed of propagation of P-waves of 6.5 km/s [46] and a maximum distance between two neighboring stations of 40 kms if the time difference between two consecutive picks of two different stations is greater than 6 s, then the peak is not associated. When n stations detect a peak and the software considers them consistent, the localization phase begins. In this case, the proposed localization algorithm based on TDoA and the hyperbola method (to be detailed in the second part of this section) is utilized. However, the architecture’s modular design offers the flexibility to potentially replace the localization module with alternative solutions that can take advantage of decentralized picking and reduced data latency. After the event is located, the system calculates the PGA from each trace on the 3 s following the detection of the P phase. If these values exceed a configurable threshold, the event is declared by publishing a JSON message on the /earthquake topic on the broker containing the collected data. At the same time, the expected arrival time of the S-waves towards one or more target points is calculated. A detailed description of the messages exchanged between the entities that constitute the proposed architecture is shown in Figure 2. Algorithm 1 describes the operations carried out to check the consistency of the detected picks and subsequently perform localization and event declaration.

The application layer is completely developed on AWS and is mainly based on an Amazon EC2 instance. EC2 is a web service that provides secure and scalable compute capacity in the cloud. The AWS SNS service, a web service that coordinates and manages the delivery or sending of messages to endpoints or to subscribing clients, is used to send a notification to higher layers when a message is published on the /earthquake topic. The cloud implementation and the use of the MQTT protocol let users easily deploy new functionalities on the applicationlayer besides the ones required for localization and event declaration, such as device monitoring services, over-the-air configuration updates, data storage, Seedlink servers, etc. Figure 3 shows a full cloud IoT architecture with the aforementioned additional services that can be deployed alongside the previously described module for detection and localization. Device monitoring, an essential feature in IoT applications, is implemented with the AWS Device Management Service (DMS). AWS DMS is fully integrated with AWS IoT Greengrass, which can be installed on the perception layer devices, with additional AWS services, such as SNS for quick notifications on device malfunctions, and the AWS Relational Database Service (RDS), for the storage of device state logs.
**Algorithm 1** Operation: Association of Arrival Times of P Phases, Localization, Event Declaration.
  1:**function** AssociateArrivalTimesOfPPhases  2:       // Data structure to store detected P phases  3:       p_phase_picks=[]  4:       // Data structure to store associated P phases  5:       associated_p_phase_picks=[]  6:       **while** len(p_phase_picks)<configurable_number_ofpicks **do**  7:             // Wait for and process incoming P phase pick messages  8:             incoming_pick=waitForIncomingPPickMessage()  9:             append(p_phase_picks,pick_info)10:       **end while**11:       // Time coincidence criterion: 6-second threshold12:       **for** each pick_info in p_phase_picks **do**13:             **for** each stored_pick_info in p_phase_picks **do**14:                   **if** pick_info.device_id≠stored_pick_info.device_id **then**15:                         time_difference=calculateTimeDifference(pick_info.time,stored_pick_info.time)16:                         **if** time_difference<6 seconds **then**17:                               // Associated with the same event18:                               append(associated_p_phase_picks,pick)19:                         **end if**20:                   **end if**21:             **end for**22:       **end for**23:       **if** len(associated_p_phase_picks)>=5 **then**24:             // Perform localization25:             localizationResults=PerformLocalization(associated_p_phase_picks)26:             **if** isEventDeclarationThresholdMet(localizationResults) **then**27:                   // Declare the event28:                   declareEvent(localizationResults)29:             **end if**30:       **end if**31:**end function**

Finally, the enterprise layer receives data and notifications from the application layer and makes them available to users through web interfaces or other applications. Interactive dashboards can be built for the visualization in real time of seismic traces if the devices are also configured as Seedlink servers. Dashboards can also be used to visualize the devices’ states using AWS RDS for continuous state monitoring.

### 3.2. Localization Algorithm

A localization algorithm based on the hyperbola method and TDoAs was developed and integrated in the previously described software module on the cloud. The hyperbola method is a graphical method for earthquake localization. Within the development of EEW systems, graphical location algorithms have, once again, become interesting since they can provide a quick and reliable location of the earthquake source [29].

The problem of position estimation from TDoA measurements occurs in a range of applications, mainly in wireless communication networks, radar, and mobile applications. It introduces many advantages compared to other localization methods such as time of arrival, including accuracy in non-line-of-sight environments, higher precision, and it removes the need of synchronizing the transmitter and the receiver [38]. In recent years, it has been also employed in earthquake localization [36,37,38], but with complex 3D or hybrid models that also focus on focal parameter determination, which is not the goal of the proposed system, since those can be estimated more accurately after the early alert. Multilateration using TDoA is based on the assumption of a constant speed of propagation. In this case, an average speed of propagation of the P-waves equal to 6.5 km/s was considered. Furthermore, observing the data relating to earthquakes that occurred in the Apennines chain of central Italy, the seismicity was included in a 15 km thick Crust Seismicity Layer (CSL) with a concentration around 10 km [47]. The aforementioned values for the average speed and depth of the earthquake were, therefore, fixed as input values for the localization algorithm. However, these values are configurable, making it is possible to test the system on different settings and different velocity models; thus, the algorithm can be modified to work in any region if the focal parameters and velocity model for that region are known.

The hyperbola method is a fairly simple graphical earthquake localization method that was developed by Mohorovičić and was frequently used until the development of more advanced computer-based methods [48]. Considering a seismic network of *n* stations, for each pair of stations j,k that detect the earthquake wavefront, it is possible to write the expression in Equation (Equation 1),
(1)$$\begin{array}{cc} v(t_{k}-t_{j}) & =d_{k}-d_{j}, \end{array}$$
where $d_{k}$ and $d_{j}$ are the distances between the epicenter and the second and first station to detect the event, respectively, $t_{k}$ and $t_{j}$ are the arrival times at each station, and *v* is the wave velocity [30,48], which is assumed to be constant. The equation shows that the possible epicenter locations are such that the difference between the epicenter distances from these two stations remains constant, meaning that the epicenter must lie on a hyperbola with the two stations as the foci. As more stations detect the earthquake wave, it is possible to determine more hyperbolas whose intersection can be used to determine a possible epicenter location. However, fixing the velocity model and average depth may lead to estimation errors if they differ from the actual values [48]. Additionally, other potential errors can arise from inaccurate data or pick times.

The algorithm developed in this work is based on the hyperbola method but it uses TDoA multilateration to build the hyperbolas, refining the epicenter estimation and reducing the error iteratively as more stations detect the P-wave and report its arrival time. The TDoA can be used to determine the distance between a radio-frequency emitter and a receiver. In this case, the epicenter plays the role of the emitter and the stations are the radio receivers. For example, assuming that the epicenter is on a circle of radius *d* meters from the station where the signal was received first, then it must also be on a circle of radius $d+V\cdot T_{DOA}$ meters from the station where the signal was received second, where *V* is the propagation speed and $T_{DOA}$ is the difference between the arrival times at the two stations. The epicenter must then lie at the intersection of the two circles, offering two possible locations. By iterating over a range of values for *d* and at each iteration finding the intersection of the two circles, a locus of possible epicenter locations can be determined. This locus forms a hyperbola on which the epicenter is bound to lie. Figure 4 shows the evolution of the developed algorithm with two stations for increasing values of *d*.

While using only two stations determines one locus of possible locations, with n>2 stations, this process can be repeated between the station that first received the message and every other station in the network/subnetwork to produce n−1 loci. As in the standard hyperbola method, finding the point with the maximum number of loci intersections then offers the best estimate of the epicenter position. The algorithm, therefore, corrects the localization estimate for each new received pick, producing the final result when the number *n* of stations that reported the P-wave is equal to a value read from a configuration file. For example, with n=4 stations, the algorithm provides n−1=3 hyperbolas. Figure 5 shows the process of constructing the hyperbolas in a four-station network by iterating over increasing values of *d*, after each station in the network detects the the P-wave. The epicenter is indicated with $T_{X}$. When the first station detects the event (S1), the algorithm places the epicenter on the station itself. After the second station (S3) picks the P-wave, the first hyperbola is obtained (red hyperbola) by iterating over different values of *d*, and the epicenter is placed at the center of the hyperbola. As a third station (S2) detects the event, it is possible to obtain a second hyperbola (green hyperbola) and a first intersection, reducing the error obtained from the first two iterations. This process can be repeated for each new station that detects the event in the network, obtaining more hyperbolas and an increasingly better estimate. The algorithm stops when a predefined number of stations detects the event, and places the epicenter on the point of maximum intersection. A detailed description of the algorithm and the hyperbola construction methods is shown in Algorithm 2.
Figure 5Hyperbola construction with the developed algorithm in a network with 4 stations named S0 to S3. The epicenter is indicated with TX. Iterating over the value of *d*, 3 hyperbolas are obtained whose point of intersection offers a good estimate of the epicenter location.
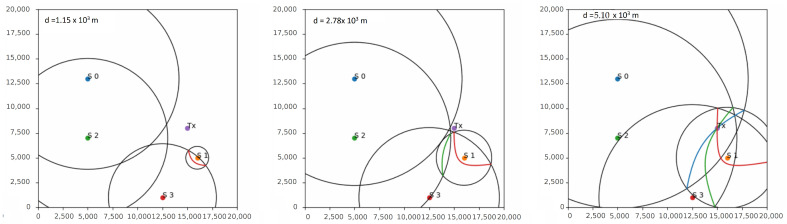

**Algorithm 2** Localization Algorithm based on Hyperbola Method and TDoA Multilateration.
  1:**function** PerformLocalization(triggeredStations)  2:      **if** len(triggeredStations)=1 and minStations=1 **then**  3:             // Place epicenter at the triggered station  4:             epicenter←triggeredStations[0]  5:      **else if** len(triggeredStations)=2 and minStations=2 **then**  6:             // Build a hyperbola and place epicenter at its center  7:             // Build a hyperbola based on TDoA  8:             hyperbola←BuildHyperbola(triggeredStations)  9:             // Get the center of the hyperbola10:             epicenter←(hyperbola.center)11:      **else**12:             // Build n-1 hyperbolas and place epicenter at max intersections point13:             epicenter←BuildMaxIntersectionsPoint(triggeredStations)14:      **end if**15:      **return** epicenter16:**end function**17:**function** BuildHyperbola(coord_k,coord_j,t_k,t_j,propagation_speed)18:      $d\leftarrow \mathrm{linspace}(1,max_{value})$19:      intersections←[]20:      **for** dist **in**
*d* **do**21:             $\mathrm{circle}\_k\leftarrow \sqrt{{\left(\mathrm{coord}\_k\left[0\right]-\mathrm{dist}\right)}^{2}+\mathrm{coord}\_k{\left[1\right]}^{2}}$22:             $\mathrm{circle}\_j\leftarrow \sqrt{{\left(\mathrm{coord}\_j\left[0\right]-\left(\mathrm{dist}+\left(t\_j-t\_k\right)\times \mathrm{propagation}\_\mathrm{speed}\right)\right)}^{2}+\mathrm{coord}\_j{\left[1\right]}^{2}}$23:             **if** circle_k+circle_j>|coord_k[0]−(dist+(t_j−t_k)×propagation_speed)| **then**24:                   $x\_intersection\leftarrow \frac{{\mathrm{circle}\_k}^{2}-{\mathrm{circle}\_j}^{2}+{\left(\mathrm{dist}+\left(t\_j-t\_k\right)\times \mathrm{propagation}\_\mathrm{speed}\right)}^{2}}{2\times (\mathrm{dist}+(t\_j-t\_k)\times \mathrm{propagation}\_\mathrm{speed})}$25:                   $y\_intersection\leftarrow \sqrt{{\mathrm{circle}\_k}^{2}-x\_intersection^{2}}$26:                   append(intersections,(x_intersection,y_intersection))27:             **end if**28:      **end for**29:      center←mean(intersections,axis=0)           ▹ Find the center of the hyperbola30:      $a\leftarrow \mathrm{mean}(\sqrt{{\left(x-\mathrm{center}\left[0\right]\right)}^{2}+{\left(y-\mathrm{center}\left[1\right]\right)}^{2}}\;\mathrm{for}\;\left(x,y\right)\;\mathrm{in}\;\mathrm{intersections})$31:      $b\leftarrow \sqrt{a^{2}-{\left(\mathrm{coord}\_k\left[0\right]-\mathrm{center}\left[0\right]\right)}^{2}}$32:      **Return** Hyperbola(center,a,b)33:**end function**34:**function** BuildMaxIntersectionsPoint(triggeredStations)35:      // Data structure to store intersections36:      intersections=[]37:      **for** i←1 to len(triggeredStations)−1 **do**38:             // Build hyperbola between triggeredStations[0] and triggeredStations[i]39:             hyperbola←BuildHyperbola(triggeredStations[0],triggeredStations[i])40:             // Calculate intersections with existing hyperbolas41:             **for** j←0 to len(intersections)−1 **do**42:                   intersection←CalculateIntersection(hyperbola,intersections[j])43:                   append(intersections,intersection)44:             **end for**45:      **end for**46:      // Find the maximum intersection point47:      maxIntersectionsPoint←FindMaximumIntersection(intersections)48:      **return** maxIntersectionsPoint49:**end function**

## 4. Performance Evaluation

In the following section, the performance of the proposed solution is evaluated through simulations on real events from Central Italy. Tests for both the localization algorithm and the overall architecture are carried out. A brief benchmarking of the AWS SNS service for notification delivery to endpoints is also carried out to test its suitability for warning dissemination once the architecture correctly declares an event. The measured metrics are error distributions and the average error for the entire testing dataset and for a strong motion subset, besides various time-related metrics such as the algorithm execution time, triggering times, and event notification delivery times.

The overall end-to-end system is compared to the Italian EEWS PRESTo, which is used as a reference to evaluate the speed of the developed system, as it represents the golden standard for regional EEW systems in Europe and Italy, as discussed in Section 2. The comparison with PRESTo is conducted by simulating a strong motion event in “playback mode” and measuring different latency-related metrics with real waveforms, as latency is one of the main constraints in EEW systems [49].

PRESTo contains the following modules [13]:Data acquisition and processing;Event detection;A localization module using the RTLoc algorithm;A magnitude estimation module using the RTMag algorithm;PGA prediction at specific targets.

In PRESTo, waveform data are collected from stations in the network using the Seedlink protocol. The event detection is carried out at a central hub using the FilterPicker algorithm. The proposed solution employs the same detection algorithm, but picking is performed directly on board the IoT devices, contrary to the usual regional EW approach. PRESTo also estimates the magnitude of the event, while this task is not carried out in the proposed system, so the comparison stops at localization.

### 4.1. Dataset

To run the simulations, a dataset was created using data from the INGV Strong Motion Database [50]. All the events with ML ≥ 3.5 that occurred from 1 January 2016 to 27 May 2020 between 42.02 and 43.82 latitude and 12.04 and 13.4 longitude were selected, with 175 events being obtained. The average depth of the events in the dataset was 9.7 km. The magnitudes ranged from 3.5 to 6.5. For each event, traces from the surrounding stations were collected, obtaining a total of more than 5000 traces. Histograms describing the magnitude and depth distribution of events in the dataset are reported in Figure 6. The geographical distribution of events and stations that detected the events are shown in Figure 7.

The overall system comparison was conducted on the 26 October 2016 17:10:36 UTC seismic event of 5.4 magnitude, with the epicenter located at the coordinates 42.879 latitude and 13.129 longitude, which is a valuable representative of an “Early Warning-grade” seismic event in this dataset.

### 4.2. Localization Algorithm Evaluation

The simulations for the evaluation of the developed localization algorithm were carried out on a machine with the following features: Intel Xeon X5650 (x2) CPU, 12 MB cache, 2.66 GHz, 16 GB RAM with Ubuntu 18.04.1 LTS. The MQTT broker Mosquitto was installed on an AWS EC2 instance and was used as the broker for algorithm evaluation. The algorithm was developed in Python and tested in simulation mode on the traces of a selected number of events using the Obspy library [51].

A tool that simulates the real-time behavior of stations and/or IoT devices during the selected events was developed. The tracks were passed to the program in an SAC format [52] or miniSEED, and analyzed at the sampling step. The picking algorithm carried out the analysis every 50 samples and, when a possible P phase was detected, an MQTT message was sent to the broker on a dedicated topic. The message contained the station data and the detection and pick timestamps. The software that performed the localization was subscribed to the same topic and began the estimation of the epicenter as soon as it received the first pick, placing the epicenter on the first station that detected the event.

Below are the detailed results of the simulation for the 26 October 2016 17:10:36 UTC seismic event of 5.4 magnitude, with an epicenter located at the coordinates 42.879 latitude and 13.129 longitude. The remaining part of this section presents the results obtained from testing the algorithm on the entire dataset. The left side of Figure 8 shows the seismograms of the first six stations that recorded the reference event, while the right side shows the execution of the algorithm after the picking of the P phase on four stations. The graph of the seismograms on the left side of the figure is ordered according to the distance from the epicenter of the event. The origin of the time axis was 17:34:04.00 UTC. After 3.45 s, the FEMA station, which was 8.259 km from the epicenter, performed the detection of the P phase and sent an MQTT message to the broker on the cloud. The subscriber thread then estimated the first localization by setting the position of the station as its epicenter. At 7.11 s, the same procedure was carried out by the GUMA station. The algorithm began the localization estimation by placing the epicenter along the hyperbola obtained, precisely at the center of the hyperbola. For each new pick that was received, the algorithm corrected the localization estimate until the number of stations from which it received P phase picks was equal to a configurable threshold. The map on the right side of Figure 8 shows the output of the algorithm after four stations detected a P phase (indicated with green triangles), while the stations that still need to detect the phase are indicated with blue triangles. The real epicentral position of the earthquake is indicated with the red star and the estimated position with the blue diamond. For completeness, the hyperbolas obtained at this point of processing are also indicated.

The time evolution of the error in meters as each station detects the P-wave is reported in Figure 9. The origin of the time axis was the instant at which the pick was detected at the FEMA station (at which time the epicenter was placed on the station). It should be noted that after receiving data from just two stations (instant t = 3.66 s), the algorithm had an error of less than 6 km. The error decreased as the information received from the stations increased, until it stabilized below 2 km after only four stations (t = 5.99 s). With a stop condition of five coherent picks for the algorithm, the final error was less than 2 km in just over 6 s.

It should be emphasized that in an EEW system, there is an evident trade-off between the correctness of the results and the system response time. An error such as the one obtained for this strong motion event after binding the phases from five stations is to be considered acceptable [13,30,49]. Furthermore, depending on the network density and the acceptable errors, the algorithm might be stopped earlier for a faster response [21].

The above simulation was repeated for all the earthquake events in the dataset. Figure 10a shows the graph of the error distributions on the complete dataset, dividing the abscissa axis into bins of 5 kms. The histograms were normalized with the relative frequencies on the total of the analyzed samples, equal to 175 events. The distribution had a positive asymmetry and an interquantile range of 8.804 km. The mean value was 9.6307 km, the median was 5.2851 km, and the 90th percentile was 22.340 km.

In Figure 11, the distribution of the execution times of the algorithm is reported, with a mean value of 163 ms, a standard deviation of 154 ms, and a 90th percentile of 209 ms. The execution time of the algorithm did not suffer due to the relatively large number of stations involved in the configuration, except for some outliers above 500 ms, which might be related to AWS EC2 slowing down due to threads or processes running in parallel to the algorithm execution. Moreover, it is an overall convenience to have a greater concentration of stations in the EEW network, since it leads to a reduction in response times, allowing the algorithm running preemptively with acceptable errors and possibly with a reduced number of triggers required for the algorithm stop condition [21].

The average depth of the events in the dataset was 9.7 km, close to the input depth chosen for the velocity model in the proposed algorithm, but there were some deviations from this average value. Figure 12 shows the deviations from the real epicentral position as a function of the magnitude and depth of the event. Depth does not have a strong correlation to high errors, also considering the close adherence to the simplified initial hypothesis of 10 km depths and the relative velocity model. Analyzing the output logs, it can also be seen that, considering only a subset of the test dataset consisting of events with a magnitude greater than 4.5, errors stayed inside an acceptable range, with the exception of some outliers above 30 km. In this reduced subset, the interquantile range was 17.4 km and the 90th percentile was 36 km. A relatively higher mean error for high magnitude events can be witnessed (13.7 km); however, this subset contained a much lower number of events compared to lower-magnitude events. Table 3 includes a summary of the error statistics for the whole dataset and for the high-magnitude subset. A summary of the execution time statistics is also reported in Table 4. Considering that an EEW system should issue alarms in the event of potentially medium/high-intensity events, and keeping in mind the looser focus on accuracy in the case of early warning for strong motion events [30], this result can be considered good for the application scenario.

### 4.3. Overall Performance Evaluation

In this section, the performance of the entire proposed system with the EEW system PRESTo using the same seismic event as outlined in the previous section (the 26 October 2016 17:10:36 UTC event, located at the coordinates 42.879 latitude and 13.129 longitude, with a magnitude of 5.4) is assessed. PRESTo can be used either in real time or offline. In real time, the system interfaces directly with the underlying seismic network, while in the second mode it operates on data files, previously acquired or generated, related to a seismic network in the SAC format. In the simulation mode, PRESTo reads SAC files and converts them into simulated SeedLink streams with 1 s data packets. It is possible to simulate network latencies and failures using configurable random gaps and delays. A simulation that is close to reality needs to use one or more SeedLink servers to stream measured ground motion towards PRESTo. In order to conduct the comparison, PRESTo is installed on the same AWS EC2 instance as the proposed application is deployed, and the SeedLink server to which PRESTo is connected is installed on the same machine as the devices and the stations are simulated. Unlike PRESTo, the proposed system does not estimate the earthquake magnitude before sending the alert; therefore, the comparison stops at localization, assuming that an alert can be sent immediately after this procedure. In addition, an assessment of the timing of the alert distribution using the AWS services described in Section 3 is included.

To conduct the simulation, the event was selected and PRESTo was configured by generating the travel-time grids of the stations using the NonLinLoc software [28,53]. The necessary configuration files were also produced, such as the description of the network and the parameters relating to automatic picking and localization. These parameters were also used in the configuration files. At the start of the simulation, the start times of the traces were modified with the current timestamp, so as to reproduce the event as if it were in progress at the time of the simulation itself. Then, a thread simulated the stations, creating a 1 s miniSEED packet and sending it to the SeedLink server to which PRESTo was connected. At the same time, each sample was placed in a queue for real-time on-board analysis for the detection of P phases by the proposed system. At the time of detection, a message was created and sent to the AWS EC2 instance via MQTT. Since both PRESTo and the proposed system use the filter picker algorithm, the results of the detection algorithm were the same. As a measure of the system’s picking performance, we quantified the time difference between the arrival time of the P-wave in a trace and its detection by the proposed system and by PRESTo.

The first phase detected was on the accelerometer track of the FEMA station. PRESTo detected the P phase after 2.13 s, while the proposed system did after 312 ms. This time difference was mainly due to the different picking modes: the simulated device can carry out the analysis directly on board, while PRESTo must wait for the filling of 1 s miniSEED packets and their transmission from the SeedLink servers installed at the station location before it is able to run the picking algorithm. Similar results were obtained on all the traces, as reported in Table 5, where the coordinates of the stations are also indicated.

The output of the localization algorithm was provided after binding the P picks from a configurable number of stations. This parameter was set to five for both systems; therefore, the detection instant of the fifth and last phase was set as the starting instant for computing the localization times. This time is referred to as $t_{trig}$, while the time to compute the localization is referred to as $t_{loc}$. PRESTo offered the first indication of localization after 1.64 s with an initial error of 13.34 km, while the proposed system offered an error of 1.341 km after 251 ms. Therefore, the total times to produce an alert for both systems were derived as follows:(2)$$\begin{array}{cc} t_{alert_{\mathrm{PRESTo}}} & =t_{trig_{\mathrm{PRESTo}}}+t_{loc_{\mathrm{PRESTo}}}=2.41\;s+1.64\;s=4.05\;s, \end{array}$$(3)$$\begin{array}{cc} t_{alert_{\mathrm{Proposed}}} & =t_{trig_{\mathrm{Proposed}}}+t_{loc_{\mathrm{Proposed}}}=0.332\;s+0.251\;s=0.583\;s. \end{array}$$

The total advantage obtained using the proposed system is, therefore,
(4)$$\begin{array}{c} t_{alert_{\mathrm{PRESTo}}}-t_{alert_{\mathrm{Proposed}}}=3.39\;s. \end{array}$$

At this point, PRESTo estimated the magnitude in less than 100 ms and calculated the expected value of the PGA towards a target point. In this case, the configured point was located in Ancona, Italy, that is, 84.87 km from the epicenter. The point of interest alert was sent when 14.49 s remained before the arrival of the S-waves, but it was not forwarded to any other type of device. Having anticipated the total detection of the event by about 3.39 s compared to PRESTo, the proposed system was able to issue the alarm when there were still 17.88 s for the arrival of the S-waves at the target point, allowing for the reduction in the so-called blind zone by approximately 11 km.

### 4.4. User Notification Performance Evaluation

Regarding the evaluation of notification times to endpoints configured in the AWS cloud, the log provided by the platform on the AWS CloudWatch service was analyzed. In particular, the CloudWatch platform was used to record the instant in which the algorithm declared the event (and, therefore, published an MQTT message) and the instant in which AWS SNS sent the notification to the configured endpoint. When an earthquake is declared, according to Figure 1, the software module running on AWS EC2 publishes a message on an MQTT broker on the topic /earthquake. A simple rule is then implemented to trigger AWS SNS when the message is published on that specific topic. The average rule execution time (i.e., the time elapsed between the *Publish-In* and that of *RuleExecution*) was equal to 143 ms, while the time for the actual delivery to an endpoint was, on average, equal to 207 ms. During this benchmark, almost 1000 message were sent with no lost messages. Both services were in the same AWS region.

## 5. Conclusions

In this work, an end-to-end cloud-IoT architecture for regional earthquake early warning is proposed and tested. The solution integrates a new earthquake source localization algorithm based on TDoA and the hyperbola method. The key novelty of the proposed system involves conducting P-wave picking directly on board IoT devices, which then transmit the gathered picking information to the cloud through the MQTT protocol. A software module on the cloud checks the temporal consistency of earthquake triggers received from the devices, performs the localization of the event, estimates the event PGA value, and sends a notification to users if the PGA exceeds a configurable threshold. Simulations show that the proposed solution is able to provide a first localization of the event with acceptable errors for the early warning use case, and to send a first alert to a target point 3.39 s earlier than PRESTo, the Italian standard of reference for regional EEW. The tests particularly highlights the benefits of moving picking nearer to the data source (in this case, the IoT devices) and communicating pick-related information through the MQTT protocol. As a result, localization and event declaration become significantly faster compared to that achieved with the conventional regional early warning systems that rely on the continuous processing of incoming Seedlink streams at a central hub. The proposed decentralized approach contributes to decreased data latency, a major factor affecting end-to-end latency in regional monitoring networks. Considering the application scenario of an EEW system, the errors obtained with the proposed algorithm (an average localization error of 9.8 km) are also to be considered admissible and in line with the use case and previous works, while also having low average execution times. The cloud integration simplifies the deployment and management of additional services that can be integrated in the proposed framework, such as data storage, device management, and direct event notification to endpoints with low delivery times.

## Figures and Tables

**Figure 1 sensors-23-08431-f001:**
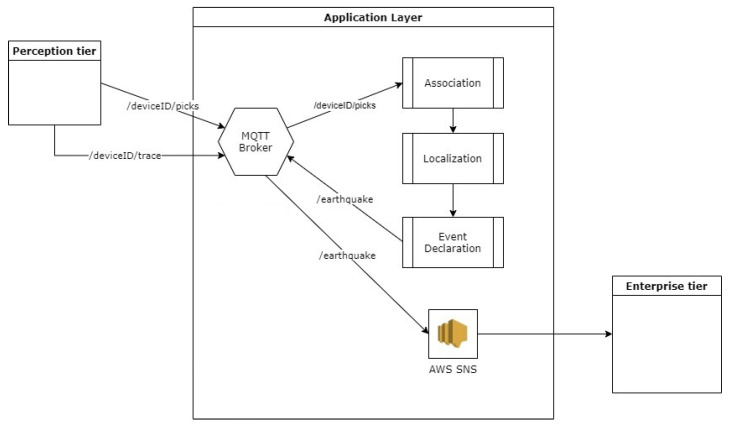
Logic flow of the localization and event declaration software module on the applicationlayer. Once a P-wave is detected, the pick information is forwarded to a broker alongside a short trace around the P-wave “pick”. The broker forwards this information to a software module on the cloud to perform pick association, localization, and, when a certain threshold is met, event declaration. AWS SNS is used to notify the event to endpoints.

**Figure 2 sensors-23-08431-f002:**
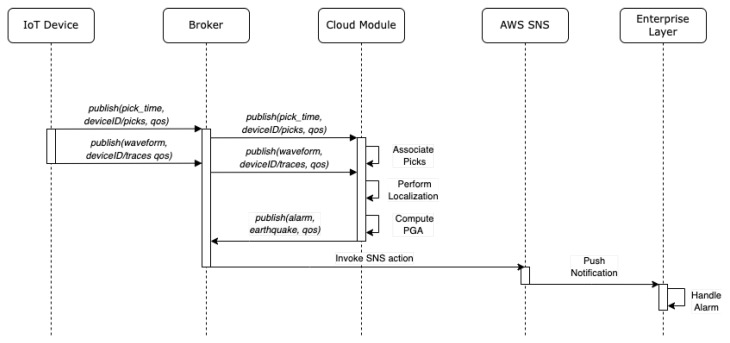
Messages exchanged between entities of the proposed architecture for localization and event declaration/notification.

**Figure 3 sensors-23-08431-f003:**
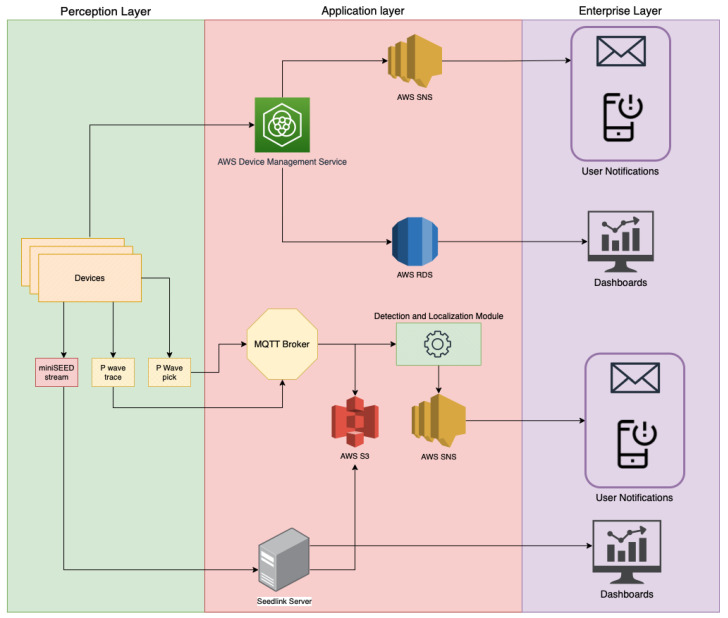
Detailed overview of the proposed cloud-IoT application, including the additional services that can be deployed alongside the localization and event declaration module.

**Figure 4 sensors-23-08431-f004:**
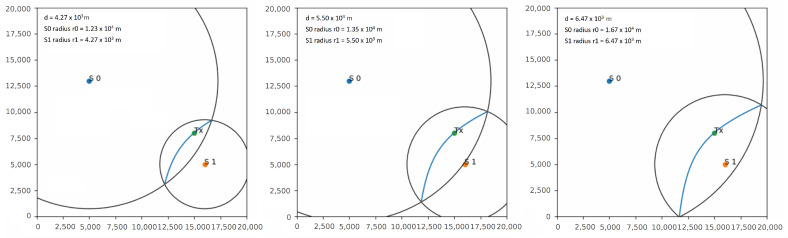
Hyperbola construction with the developed algorithm in a network with 2 stations called S0 and S1. The epicenter is indicated with $T_{X}$. Iterating over different values of the radius of the circle centered around S1 (*d* parameter), a locus given by the intersection of the two circles is obtained, on which the epicenter is bound to lie.

**Figure 6 sensors-23-08431-f006:**
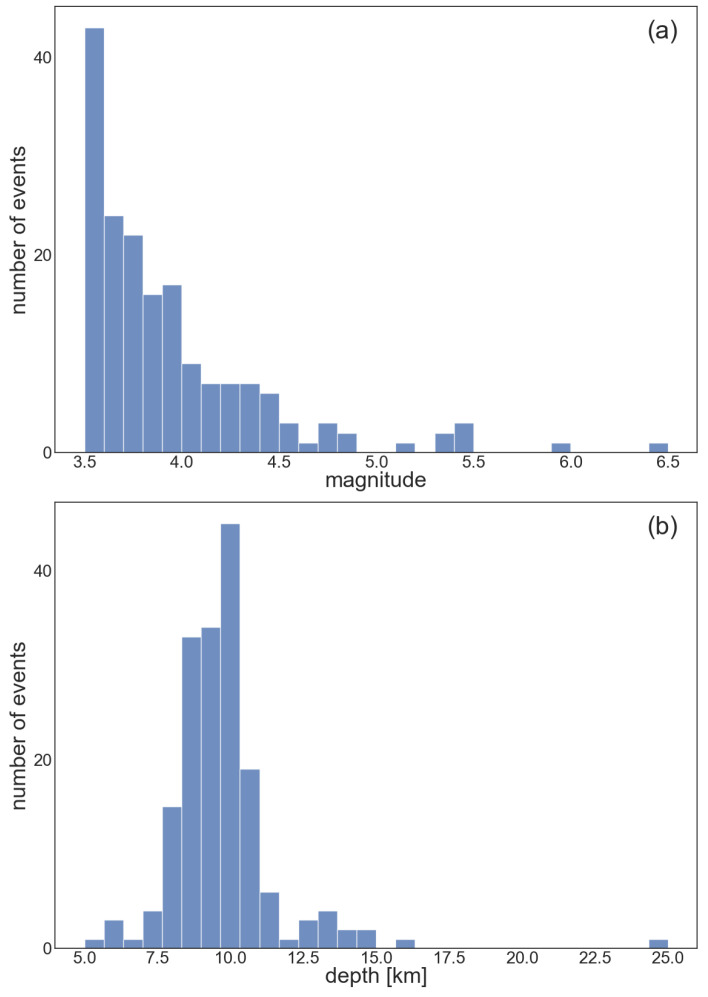
Histograms of the magnitude (**a**) and depth (**b**) distributions in the dataset.

**Figure 7 sensors-23-08431-f007:**
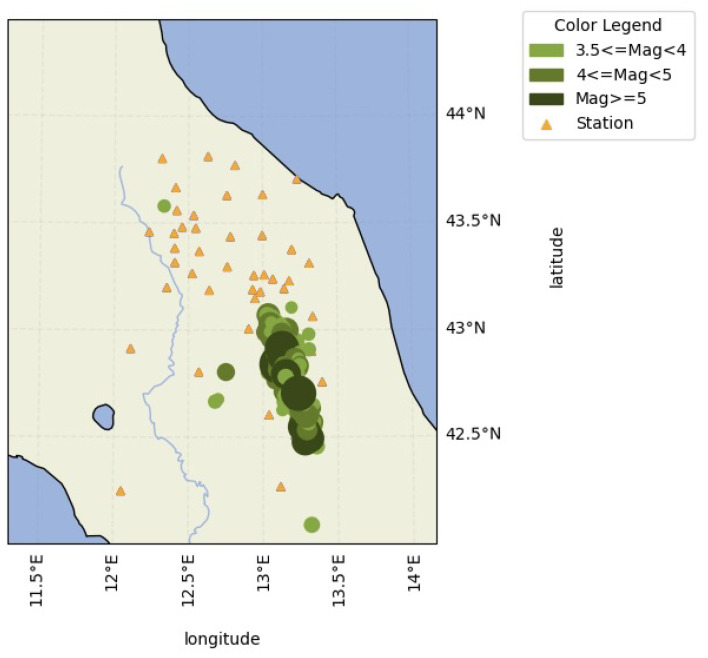
Map reporting the events (circles) in the dataset and the stations (triangles) that detected the events. The size of the circles denotes the magnitude.

**Figure 8 sensors-23-08431-f008:**
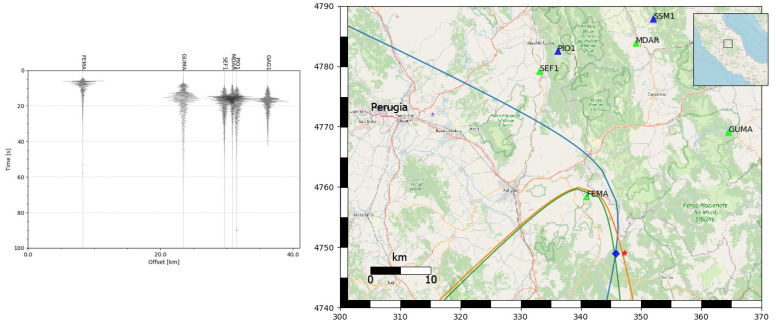
Seismograms of the first 6 stations that recorded the 26 October 2016 17:10:36 UTC event, located at the coordinates 42.879 latitude and 13.129 longitude, ordered according to the distance from the real epicentral position (on the **left**), and the execution of the localization algorithm (on the **right**). The red star indicates the real epicenter, the blue diamond the estimated one.

**Figure 9 sensors-23-08431-f009:**
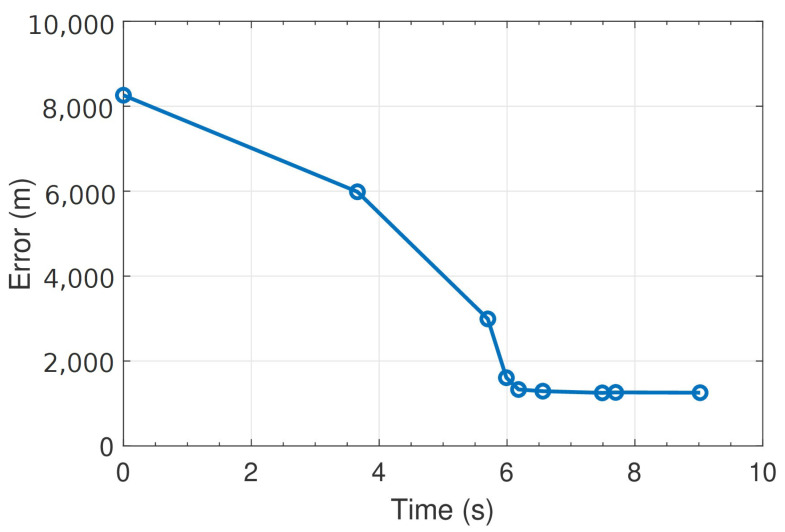
Time evolution of the error in meters as more stations detect the event. The instants corresponding to the algorithm receiving a new pick are indicated with the blue circles.

**Figure 10 sensors-23-08431-f010:**
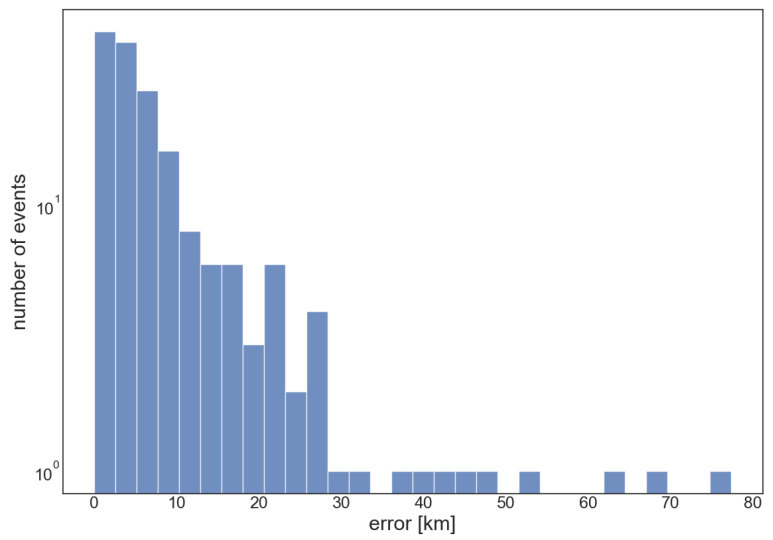
Errors distribution. The abscissa bins are 2.5 km wide.

**Figure 11 sensors-23-08431-f011:**
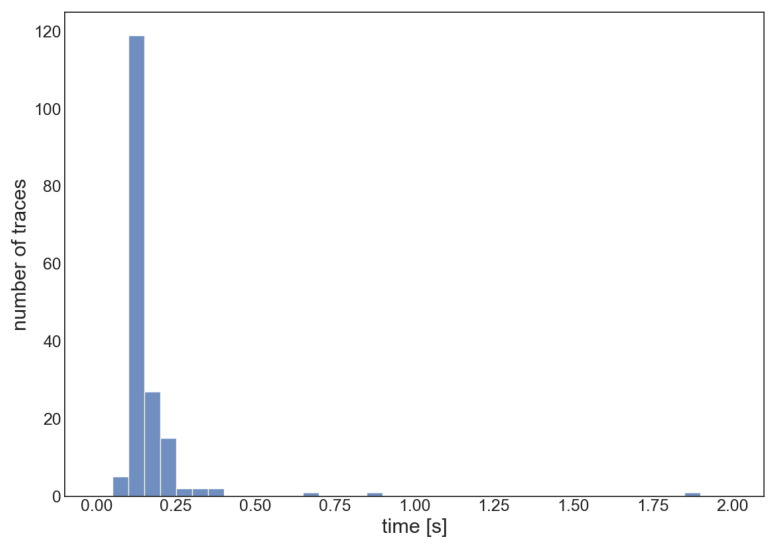
Execution times for the localization of each event in the dataset. The abscissa bins are 50 milliseconds wide.

**Figure 12 sensors-23-08431-f012:**
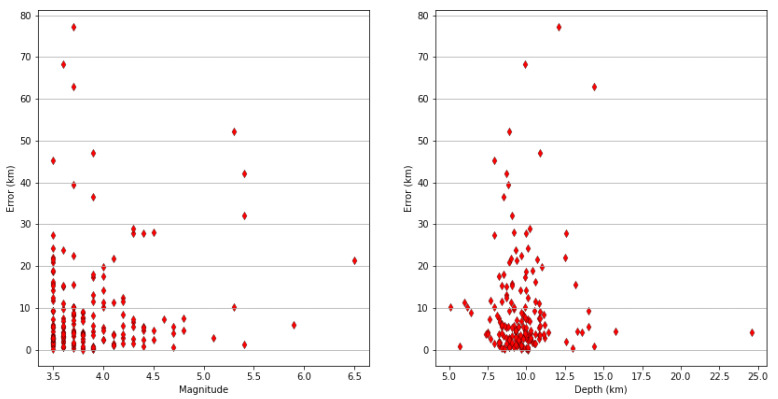
Error distributions as a function of the event magnitude (on the **left**) and depth (on the **right**). The red diamonds represent errors in kilometers corresponding to specific magnitudes (**left**) or depths (**right**).

**Table 1 sensors-23-08431-t001:** Comparison of the main features of state-of-the-art regional earthquake early warning systems.

Feature	VS(SC)	PRESTo	Proposed Solution
Picking	Centralized	Centralized	Distributed
Data Transmission Protocols	Seedlink	Seedlink	MQTT (Seedlink optional)
Picking Algorithm	Short-Time Average over Long-Time Average	FilterPicker	FilterPicker
Localization Algorithm	scautoloc	RTLoc	TDoA-based method
Magnitude Estimation	scvsmag module in Seiscomp	RTMag	None
Minimum Number of Stations	Configurable (2 for general area estimation, minimum 3 for unique point estimation)	Configurable	Configurable

**Table 2 sensors-23-08431-t002:** Comparison of the related works.

Work	Main Focus	Method for Localization	Cloud Platform	Main Novelty
[39]	Mobile Crowdsensing Architecture	Hypoinverse	MyShake Servers	Harnessing of smartphones to build a global seismic network
[26]	Cloud Architecture	None explicitly proposed	Amazon Web Services	Connected geophone network
[25]	Mobile Crowdsensing Achitecture	None explicitly proposed	Seismocloud Server	Harnessing of smartphones and sensors to build a “community” EW network
[35]	Localization Algorithm	Random Forest based on p-pick triggers	Not specified	RF algorithm is quick and can learn from a limited amount of data
[33]	Detection and Localization Algorithm	CNN based on waveform analysis	Not specified	Joint CNN for detection and localization
[34]	Localization and Origin Time Estimation Algorithm	Attention-based hypocenter estimator based on p-pick triggers	Not specified	High accuracy in locating hypocenters and also estimating origin times of earthquakes
[32]	Localization Algorithm	Bayesian convolutional neural network based on single-statio waveforms	Not specified	Single station localization estimate
Proposed Solution	Cloud architecture and Localization Algorithm	TDoA based on p-pick triggers	Amazon Web Services	End-to-end IoT architecture and quick localization method

**Table 3 sensors-23-08431-t003:** Error Distribution and Subset Statistics.

Metric	Full Dataset	Reduced Subset (Magnitude > 4.5)
Interquantile Range	8.804 km	17.4 km
Mean Value	9.6307 km	13.7 km
Median	5.2851 km	5.9 km
90th Percentile	22.340 km	36 km

**Table 4 sensors-23-08431-t004:** Execution Time Statistics.

Metric	Value (Seconds)
Mean Value	0.16
Minimum	0.09
90th Percentile	0.21
Interquantile Range	0.04
Median	0.13
90th Percentile	0.21
Maximum	1.87

**Table 5 sensors-23-08431-t005:** Simulation results: triggering times for PRESTo and the proposed system. Times are in seconds.

Station	Lat(N)	Long(E)	PRESTo	Proposed Solution
FEMA	42.9621	13.0497	2.13	0.312
GUMA	43.0627	13.3335	1.84	0.286
SEF1	43.1468	12.9476	2.04	0.301
MDAR	43.1927	13.1427	2.41	0.410
GAG1	43.238	13.0674	2.33	0.332

## Data Availability

Not applicable.

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
