# Peer review of "A Cloud-IoT Architecture for Latency-Aware Localization in Earthquake Early Warning"

_sensors, 2023, doi:10.3390/s23208431_

Round 1
Reviewer 1 Report
The paper addresses a very intesting topic which is Localization in Earthquake Early Warning.
The topic is very relevant, the scientific soundness is good, and methodology is valid. Far better than many papers that I had the chance to read on this topic.
The quality of the presentation is high, the paper is well organised and well written. This makes easy to follow the flow and appreciate authors contribution.
The proposed architectural and methodological framework offers very promising results as well.
Given the above, I recommend the acceptance of this paper.
It would be nice if the authors could provide in the paper some insights on some questions could raise in readers of the paper (just commenting, not from a numerical viewpoint of course) following.
The targeted application is a fine grained localization of the earthquake which in state of the art is addressed in the seconds timescale. The proposed approach, however brings the elaboration time in the order of hundreds of milliseconds. This makes it close to 5G/6G and edge computing high reliability and low latency targeted applications. In this perspective, how would availability of cloud resources at the edge impact the proposed approach? Also, given the eew scenario, how unavailability of infrastructure elements (network or computation) affects the effectiveness of the proposed approach? Please see the work published on this venue:
Franchi, Fabio, et al. "What can 5g do for public safety? structural health monitoring and earthquake early warning scenarios." Sensors 22.8 (2022): 3020.
Reviewer 2 Report
The Authors propose a Cloud-IoT based decentralized architecture to improve the latency in identifying the localization of an earthquake epicentre. The Cloud-IoT proof of concept is here demonstrated and compared with traditional tools and approaches.
From my point of view, the paper requires Minor Revision.
Few comments in the following:
In the introduction it would be nice to see what is a suitable IoT device available on the market that could do the job. Would a smartphone be suitable? A raspberry py or something similar? This is to say, how much this architecture is realistically deployable? What feature an hardware should have?
Line 29: please use the EEW acronym already introduced
Line 246: topicfor
Line 280: Authors may expand here to explain very briefly how do software assess the consistency of the peaks
Reviewer 3 Report
The authors forecast earthquake using a Cloud-IoT Architecture for Latency-Aware Localization.
The approach is good & has some contributions but still need some minor corrections to improve quality.
The abstract sums up the study's aims, methodology, and results concisely. A better organized presentation, though, would improve the piece. You may want to break it up into parts devoted to the study's background, procedures, findings, and implications.
Identify the void that this research seeks to fill in the current body of knowledge. How would your study add to what we already know?
Specific machine learning algorithms and feature engineering approaches used in the research should be described. Describe the attributes that were evaluated and the thought process behind which ones were picked.
The performance of the classification model may be evaluated using a variety of assessment indicators, which should be detailed. Describe the measures like accuracy, precision, recall, and F1-score that were used to verify the model's efficacy.
Include pseudo code of your model and explain in depth.
Write main contributions at end of the section 2 after highlighting the gap to be addressed..
Check the abstract for typos and make sure it makes sense.
Figure 1 is not cited in text. Please cite it to suitable place.
There is only one table & I suggest to add min two more tables one for Performance metrics and 3rd for results comparisons in the state of art.
Use same dataset for all comparisons & also include one section for the "dataset"
Seems good only minor proof reading required.
Reviewer 4 Report
The article is well presented, and the context is of great interest. The novelty of the work is clear; however, I have the following concerns.
- All acronyms and initialisms should be defined the first time they appear in the text. This should be in the abstract as well as the article, e.g., IoT.
- Capitalization of the first letter should only be applied to proper names, but not to common names, with a few exceptions (e.g., titles).
- The introduction section is too long; reconsider the section trying to remove unnecessary paragraphs and redundant data.
- The novelty is clearly presented; however, the main contributions are not clear. Consider listing the main contribution at the end of the introduction section.
- Consider adding a comparison table at the end of the related work section to give better meaning and representation of the section.
- The authors should consider the distributed computing paradigm as a part of the system structure.
The article requires quick proofreading
Reviewer 5 Report
The authors of this manuscript present "A Cloud-IoT Architecture for Latency-Aware Localization in Earthquake Early Warning". However, the following suggestions are made to improve the readability of the manuscript.
1. Introduction should be concise and precise. Raise the research questions in introduction section. Highlight the authors contributions.
2. Related works are weak. Show the related works in a table and find the gap. Show the presented work strengths and how it is importent.
3. Line 189, can you add an algorithm in this section to show the working process?
4. Figure 1 shows very basic architecture. Modify it.
5. In Localization Algorithm section, there is no algorithm presented.
6. Add a number in front of equation in line 340. Can you add reference for this equation.
7. How are the simulation results: triggering times for PRESTo and the proposed system found in line 534. Write mathematical formula or algorithms.
Moderate editing of English language required.
Round 2
Reviewer 3 Report
All desired corrections are well inserted and so pleased to accept this work in its present form.
Seems acceptable
Reviewer 4 Report
All comments have been addressed.
The article requires a quick proofreading.
Reviewer 5 Report
The authors have revised this manuscript according to my previous comments, However, the related works section is still weak. Can you explore all related studies in a table.
Minor editing is required.
